# An ASIP for Neural Network Inference on Embedded Devices with 99% PE Utilization and 100% Memory Hidden under Low Silicon Cost

**DOI:** 10.3390/s22103841

**Published:** 2022-05-19

**Authors:** Muxuan Gao, He Chen, Dake Liu

**Affiliations:** School of Information and Electronics, Beijing Institute of Technology, Beijing 100811, China; gao.muxuan@foxmail.com (M.G.); dake@bit.edu.cn (D.L.)

**Keywords:** deep neural networks, machine learning, deep learning processor, scheduling framework, instruction set architecture (ISA)

## Abstract

The computation efficiency and flexibility of the accelerator hinder deep neural network (DNN) implementation in embedded applications. Although there are many publications on deep neural network (DNN) processors, there is still much room for deep optimization to further improve results. Multiple dimensions must be simultaneously considered when designing a DNN processor to reach the performance limit of the architecture, including architecture decision, flexibility, energy efficiency, and silicon cost minimization. Flexibility is defined as the ability to support as many multiple networks as possible and to easily adjust the scale. For energy efficiency, there are huge opportunities for power efficiency optimization, which involves access minimization and memory latency minimization based on on-chip memory minimization. Therefore, this work focused on low-power and low-latency data access with minimized silicon cost. This research was implemented based on an ASIP (application specific instruction set processor) in which an ISA was based on the caffe2 inference operator and the hardware design was based on a single instruction multiple data (SIMD) architecture. The scalability and system performance of our SoC extension scheme were demonstrated. The VLIW was used to execute multiple instructions in parallel. All costs for data access time were thus eliminated for the convolution layer. Finally, the processor was synthesized based on TSMC 65 nm technology with a 200 MHz clock, and the Soc extension scheme was analyzed in an experimental model. Our design was tested on several typical neural networks, achieving 196 GOPS at 200 MHz and 241 GOPS/W on the VGG16Net and AlexNet.

## 1. Introduction

Neural networks have evolved and become deeper and larger over the last decade. Due to training based on very large datasets, deep neural networks have a degree of accuracy vastly superior to that achieved using any other AI technologies in many areas, such as image recognition, natural language processing, text analysis, etc. [1,2,3]. Since accuracy is no longer an obstacle, deep neural network (DNN)-based AI technologies have become feasible for many applications. However, another challenge appears when implementing DNNs in embedded applications of Internet of Things (IoT) devices, namely computation efficiency. DNNs are compute-intensive and data-intensive algorithms that need to be executed as fast as possible while keeping power consumption as low as possible. To achieve this, none of the available CPUs and GPUs are appropriate choices. This is because the CPUs cannot execute DNNs rapidly, whereas GPUs consume excessive power. Thus, there is a need for a customized programmable accelerator for DNNs [4].

Many customized accelerators for DNNs have been developed. For convolutional neural networks (CNNs), Yu-Hsin Chen developed Eyeriss [5], Angshuman Parashar developed SCNN [6], Boming Huang developed IECA [7], and Tu Fengbin developed Evolver [8]. Arfan Ghani proposed a hardware-based acceleration method and implemented it on FPGA for accelerating the diagnosis of novel coronavirus (COVID-19) [9]. For long short-term memory (LSTM), which is a kind of RNN, Song Han developed ESE [10], Peng Ouyang developed a specific architecture to parallel LSTM in [11], and Deepak Kadetotad also developed an accelerator to optimize the memory efficiency for LSTM [12].

All of the abovementioned accelerators could achieve excellent performance for the specific deep neural network type they were designed for. However, different types of deep neural networks are often merged together to achieve specific functions in real applications; for example, Yin Fan proposed video-based emotion recognition using CNN-RNN and C3D hybrid networks [13] and Vahid Azimirad proposed a consecutive hybrid spiking-convolutional (CHSC) neural controller by integrating CNN and spiking neural networks (SNNs) [14]. Thus, flexibility is also an important aspect of accelerator design. A reconfigurable heterogeneous array-based architecture was proposed by Shouyi Yin in [15]. In this, there were two types of PEs: a general PE for convolution and full-connection, and a super PE for the pooling layer and the RNN. Each PE was reconfigured by a 12-bit configuration word. Because the reconfiguration word was limited and the PEs were designed for certain layers, the flexibility was still insufficient. In contrast to the reconfigurable architecture in [15], Shaoli Liu in [16] provided an instruction-based accelerator, Cambricon. By comprehensively analyzing existing deep neural networks, they provided an instruction set, including scalar, vector, matrix, logical, data transfer, and control instructions. Meanwhile, an accelerator was co-designed to implement this instruction set with low power consumption. By programming the accelerator with these instructions, the Cambricon could provide much more flexibility than other DNN accelerators.

From the above analysis, we conclude that the instruction-based accelerators, such as Cambricon, are power efficient and operation flexible, which is appropriate for DNN applications. Yet, there are still some aspects of Cambricon that can be improved. Firstly, Cambricon chooses ten representative, yet distinct, neural networks for its design space. Although ten neural networks could cover most of the operations, it is still not a convictive design space. Secondly, Cambricon implements the vector operation and the matrix operation with two separate functional units. However, the vector operation and the matrix operation have great similarities, which means that there are opportunities to merge these two function units together. Thirdly, the data should be loaded into the register file first, prior to computation, similar to the RISC instruction set. Although the microarchitecture can be simplified in this kind of design, the clock cycle is wasted. Improving power efficiency and processing speed are also among the most important design objectives. Hence, there is still room to improve the data preparation strategy in Cambricon.

In this paper, we propose a novel instruction set architecture (ISA) and co-optimized micro-architecture called the deep learning processor (DLP). Based on comparisons, our design outperforms general CNN accelerators for the following three reasons:(1)DLP can flexibly accelerate most CNNs and even common mathematical operations by designing a dedicated ISA and hardware system based on SIMD architecture.(2)DLP can be easily expanded at the Soc-level to increase computing power for various embedded applications.(3)Based on the hardware system, a scheduling framework is proposed to reduce the latency of memory access, improve utilization and reduce data access, which can effectively optimize performance and energy consumption.

The remainder of the paper is organized as follows: Section 2 describes the prerequisites for designing the ISA. Section 3 presents an overview of the ISA and the program examples for typical layers in the CNN. Section 4 introduces the hardware design of the DLP and the Soc-level extension scheme. Section 5 presents the details of the scheduling framework. Section 6 provides the experimental results. Section 7 concludes the whole paper.

## 2. ISA Design Considerations

To design a flexible, reduced, and efficient instruction set architecture (ISA), several aspects should be considered, such as the design space, data parallelization, approximation, and the memory subsystem [17].

### 2.1. Design Space Exploration

New DNNs are emerging so rapidly that it is barely possible to keep track of them. Fortunately, most of the DNNs are implemented on deep learning frameworks, such as tensorflow and caffe2. With the optimized operators provided by the deep learning framework, algorithm researchers could pay more attention to their innovation point rather than waste the majority of time on recoding the basic functions. Furthermore, for DNN hardware accelerator designers, the deep learning framework provides an intermediate representation gapping the algorithms from the hardware. Thus, the relatively stable deep learning framework, rather than diverse algorithms, could be chosen as the design space. By sufficiently supporting the operators in a deep learning framework, the accelerator can naturally support the majority of DNNs. We chose caffe2 as our design space, since it is optimized for mobile integration and amenable to supporting specific accelerators.

More than 500 operators are supported by caffe2 [18] and can be classified into several types: inference operators, training operators, and auxiliary operators. Here, we concern ourselves with inference operators. The operators for inference can be divided into several categories (see Figure 1): learnable layer operators, activation layer operators, normalization layer operators, pooling layer operators, output layer operators, and mathematical operators. The first three categories correspond to certain deep learning layers, whereas the last includes common mathematical tensor operations, such as addition, accumulation, dot product, logic operation, etc.

The input of these operators is always tensors with up to four dimensions. The algorithm kernels of these operators can be further divided into three types: slide-window, matrix-vector, and element-wise. Slide-window type means that there is a window restraining the operated data and the window slides across the tensor. Convolution and normalization operators belong to the slide-window type. Matrix-vector type means a matrix multiply vector. This operation is popular for full-connect layer operators, LSTM operators, and GRU operators. The matrix multiply matrix can also reduce to a matrix-vector type. Element-wise type means the same operation is performed on every element of the tensor. The activation operators, softmax operators, and most of the basic mathematical functions, belong to the element-wise type.

Before designing the ISA, the kernels should be analyzed to explore the similarities and fused to simplify the ISA. We found that all of the three kernel types could be fused and mapped to the framework shown in Figure 2a. The pre-function(i,j,k,l,m,n) and post-function(i,j,k,l,m,n) could be at the beginning and end of any one of the loops. The pre-function(i,j,k,l,m,n) is usually used to reset the temporary variable or prepare data for function, and the post-function(i,j,k,l,m,n) is usually used to restore the computation output. The function(i,j,k,l,m,n) implements the mathematical operations, such as the multiply and accumulate (MAC) operation for the convolution operator, maximum operation for the normalization operator, and sigmoid operation for the activation operator. There are six nested loops in Figure 2a, which fit the slide-window type of kernels. For other kernel types, there may not be that many nested loops. Table 1 shows the framework components for different kernel types.

According to the inference operators and the framework proposed above, we analyzed the following aspects to extract a high efficiency ISA.

### 2.2. Data Parallelization

The essence of acceleration is computing parallelization. By choosing several loop steps and simultaneously mapping them into the accelerator, the loop iteration number is reduced and the algorithm is accelerated. We mainly discuss the parallelization for convolution, which is the most complex algorithm. We also found that other algorithms could fit the parallelization strategy for convolution precisely.

Some of the CNN accelerators map the convolution across the height and width dimensions of the data tensor into a 2-D MAC array. The objective of this kind of 2-D parallelization is to utilize the data reuse opportunities in the height-width plane as much as possible. This is because when sliding the filter window across the height-width plane of the input tensor, the overlapped data can be reused. However, this kind of parallelization has poor flexibility. Since the window size and stride are varied sharply for different CNNs, and all these variations should be implemented on the 2-D MAC array, it is hard to design control and data sharing mechanisms across the 2-D PE plane with sufficient flexibility. Furthermore, this kind of 2-D MAC array is also hard to adapt to other algorithms, such as full connect and activation.

In contrast to the 2-D MAC array, the 1-D MAC array is more common in processors and is usually referred to as an SIMD architecture. By carefully choosing the mapping dimension of the data tensor, we find that, not only are flexibility problems that occurred in a 2-D MAC array eliminated, but the data can also be reused as much as possible.

We parallel the convolution operation across the channel dimension of the input tensor and the batch dimension of the filter tensor. The pseudocode of our parallel scheme is shown in Figure 2b, in which the input tensor with size $N\times H_{i}\times K_{i}$ convolves with the filter tensor of size M×N×K×K and produces an output tensor of size $M_{o}\times H_{o}\times L_{o}$. We first segment the iteration loops for TM and TN with step Tm and Tn first. Then, we take TM data from the channel dimension of each of the TM batches of the filter tensor. After that, the convolutions for these data are computed. Figure 3 shows the details of the data parallelism. The loop tm and tn in Figure 2b are unrolled onto the 1-D ALU array. A total of TM×TN operations are simultaneously performed on SIMD, so TM×TN must be less than the SIMD lane number.

For this kind of 1-D parallelization, there are several advantages. Firstly, the computation of each paralleled lane is independent. The only interaction is the summation of the lane results. This can be achieved using a triangular accumulator, which is common in SIMD micro-architecture.

Secondly, the control of each lane is uniform and consistent. This is because the cases that cause control divergence in 2-D parallelization, such as window size, stride, padding, and dilation, all occur in the loop sweep for the height and width dimensions. When we parallel the computation in the input channel and filter batch dimensions, these variations are the same for every SIMD lane. The window size is reflected in the loop iteration number of filter height and width and the stride is reflected in the coefficient of the index calculation for the input tensor. Thus, there is no control divergence among lanes and the control logic can be simplified. Because of the above analysis, the flexibility problems caused by window size and stride in 2-D mapping are eliminated. Furthermore, since the 1-D SIMD is a common architecture, most other operations can be easily implemented using the same architecture.

Thirdly, by carefully selecting TM and TN, the data reuse rate for our 1-D parallelization can even surpass that for 2-D parallelization. Although the overlapped data in the height and width dimensions of the input tensor can not be reused, our 1-D parallelization reuses the input data from the filter batch dimension. The TN data from the channel dimension of the input tensor, would compute a convolution with TN data from the channel dimension of each of the TM bathes of the filter tensor. Thus, every TN data value in the input tensor can be reused TM times. In this way, the input tensor can be reused in different MAC operations and calculated using different filter tensors.

All the above discussions are about convolution, but most of the other kernels can also be computed by a 1-D SIMD in parallel and exhibit good performance. For pooling kernels, we can parallel the computation along the channel dimension of the input tensor. Thus, similar to the convolution, the pooling window variation along the height and width dimensions does not impact the consistency of the computation of the SIMD lanes. The element-wise type of kernel can be paralleled along any dimension of the input tensor. Since the channel number of the input tensor is always a power of two and the SIMD lane number is also usually a power of two, the SIMD utilization is higher when paralleling the computation along channel dimensions rather than along other dimensions. For the matrix-vector type of kernels, it is preferable to parallel along the height dimension of the input tensor such that the element of the vector can be reused.

### 2.3. Approximation

There are some transcendental functions in activation layer operators, such as sigmoid, sin, tanh, etc. All require complex hardware to acquire precise results, which is unaffordable for mobile and embedded applications. Since DNNs can tolerate certain numerical errors [19], methods such as segmented function, Taylor series expansion, and CORDIC algorithm can be used to approximate these transcendental functions. Table 2 shows the typical transcendental functions and their approximated functions. These functions can all be implemented by basic mathematical operations with condition judging.

### 2.4. Control Acceleration

Figure 2a shows that there are several nested loops in many algorithm kernels. For traditional ISAs, the loop is implemented by a set of instructions, such as the assembly code shown in Figure 4. We can see that there are five instructions associated with the loop control. When several loops are nested and the operation instructions are limited, these loop control instructions occupy many cycles and decrease the time utilization of the processor. Hence, there should be a specific repeat instruction to accelerate the nested loops. We define time utilization as the computation code divided by the overall executed code, assuming that one repeat instruction could achieve the function of one loop. Figure 5 shows the time utilization comparison between the nested loop code with/without a specific repeat instruction. We can see that a repeat instruction can significantly enhance the time utilization ratio.

In this section, we first take the inference operators in caffe2 as our design space. Then, we discuss the data parallelization mechanism, in which all the algorithm kernels are efficiently mapped to a 1-D SIMD computing unit. After that, we approximate the transcendental functions using a segment function and Taylor expansion, which can be implemented by basic mathematical operators. Lastly, we find that the nested loops in kernels would reduce the time utilization. Hence, we add a loop operator, and a repeat, in our design space. Based on all the above discussion, we obtain a more detailed design space for hardware design in Figure 1, which includes one control operator, a repeat, and several basic mathematical operators.

## 3. ISA Overview

According to the design space we established in the last section, we propose our ISA. The overview of our ISA is shown in Table 3. The ISA is built for a customized 32-lane SIMD datapath. To simplify the instruction decoder logic, the instructions are arranged into four types of formations shown in Table 4. Except for the common segments, such as opcode, dst, src0 and src1, we add two customized segments: condition (cdt) and option (opt) (see Table 5). The condition segment is used to declare whether the instruction is conditionally executed and what the condition is. The option segment declares whether the customized functions, such as shift, round, and saturation, are executed in this instruction. For the operands, dst, src0 and src1, we define four kinds of operands in Table 6: register, scratchpad memory(SPM) with register-indirect addressing, immediate, and accumulate register. In addition, we add two customized segments for operands: length and pattern. The length indicates the length of the operand to be executed. The pattern indicates the length of the operand extracted from the SPM. When the pattern is less than the length, the extracted data would be tilted length/pattern times, by which the reused operand is implemented. This is useful for convolution in achieving input tensor reuse. Furthermore, the length segment could also control the executed SIMD lanes, through which our ISA could not only implement the vector operation with variable length, but also the scalar operation. The scalar operation is convenient for control operations and register configuration.

### 3.1. Instruction Resume

Our ISA contains five types of instructions: computation, logic, shift, control, and data transfer. The majority of the computation, logic, and shift instructions are executed on SIMD lanes in parallel. The function of most of these instructions is similar to their corresponding RISC instructions. Exceptionally, the triangular accumulate instruction (tacc) is used to accumulate dst operands of several SIMD lanes together, which is necessary for convolution.

For the control instruction, we define three traditional instructions (not, car, and jump) and one customized instruction (repeat). The jump instruction combined with the condition segment can achieve both unconditional and conditional jump operation. The customized repeat instruction is used to simplify the nested loops. The operands dst, src0, and src1 in the repeat instruction have specific meanings. They indicate the position of the repeat instruction in the nested loops, repeat number, and number of the following instructions this repeat instruction has covered.

The data transfer instructions are defined for register updating. The copy instruction moves the data between the registers, and the ldi instruction loads the immediate operand into the register. The SPM is prepared for computation, logic, and shift instructions, whose data read and write is implicit in the instruction operands.

Meanwhile, in order to convert our dedicated ISA instructions into a binary code that is executable by DLP, we designed an assembler through an open-source YACC library in Python.

### 3.2. Program Examples

To illustrate the usage of our proposed ISA, we programmed three representative layer operators in DNNs, a convolution layer, a pooling layer, and a full connect layer, which are shown in Algorithm  1.
**Algorithm 1:** Program examples**Convolution code**    ldi g0 1    ldi g1 2    ldi g2 32    ldi g3 3    ldi g4 3    ldi lacr2 0    loop5: ldi lacr1 0    loop4: ldi lacr0 0    loop3: car    repeat 2 3 4        repeat 1 2 3            repeat 0 1 3                mac acr.vi [0].ii.d [0].v.h    tacc<shl15> [0].v acr.ii.d    add lacr0 lacr0.i.v g1.i.w    sub g8 lacr0.i.v g2.i.w    jmp.ne loop3    add lacr1 lacr1.i.v g0.i.w    sub g8 lacr1.i.v g3.i.w    jmp.ne loop4    add lacr2 lacr2.i.v g0.i.w    sub g8 lacr2.i.v g4.i.w    jmp.ne loop5**Maximum pooling code**    ldi g0 1    ldi g1 2    ldi g2 32    ldi g3 3    ldi lacr1 0    loop4: ldi lacr0 0    loop3:    repeat 2 5 4        car        repeat 1 2 3            repeat 0 1 3                max acr.vi [0].vi.w        tacc<shl15> [0].v acr.vi.w    add lacr0 lacr0.i.v g1.i.w    sub g8 lacr0.i.v g2.i.w    jmp.ne loop3    add lacr1 lacr1.i.v g0.i.w    sub g8 lacr1.i.v g3.i.w    jmp.ne loop4**full connect code**    repeat 1 3 3        car        repeat 0 1 3            mac acr.vi [0].vi.v [0].vi.w        tacc<shl15> [0].v acr.vi.w

## 4. The Hardware Design

For CNN acceleration, we propose a hardware design that includes DLP, based on the instruction set of the previous section and the Soc-level scheme for computing power expansion.

### 4.1. DLP Micro-Architecture Implementation

The micro-architecture of our DLP is presented in this section, which is shown in Figure 6a. Our DLP is composed of seven major pipeline stages: instruction fetching, instruction decoding/loop accelerating, register/SPM reading, front permutation, execution, back permutation, and register/SPM writing. To simplify the architecture, we apply the in-order execution pattern in our DLP. The loop accelerator is used to achieve the repeat instruction. The SPM is actually an SRAM, and it is used to store the tensor data. Every operand of src0, src1, and dst has its private SPM, which is independently addressed. The operands for the vector instructions are vectors with up to 32 elements which need to be simultaneously provided. Thus, the SPM is composed of 32 banks, which means that each SPM can simultaneously load/store a maximum of 32 operands. We insert the SPM reading stage for the src0 and src1 and SPM writing stage for dst into the DLP pipeline, by which the data read/write process can be hidden into the computing process. Although SPM reading/writing back would extend the pipeline and increase the latency, it is affordable since the latency caused by a long pipeline is ignorable compared with the latency caused by a separate load/store instruction. In order to make the data read/write flexible, we also add two permutation stages to reorder the data. In general, DNNs can tolerate numerical errors. For neural network inference, 8-bit or even 4-bit fixed-point computing units are enough to perform as well as floating-point numbers [20]. Thus, the ALU is an SIMD structure with a 32 8-bit fixed-point MAC. All the functions of the algorithm instructions are merged into one single ALU architecture in which the silicon utilization ratio is high and the control strategy is simple.

### 4.2. Extension of the DLP Compute Capability

From now on, we describe our DLP with 32 lane SIMD architecture, whose compute capability is limited, and needs to be extended according to the DNN applications. We propose the SoC-level extension scheme, which is shown in Figure 6b. The master processor and 16 DLP can get together to compose a heterogeneous multi-core architecture. The master processor has custom VLIW instructions that can dispatch the computation of the layers into different DLP cores and control direct memory access (DMA) to transfer data. Each DLP has six independent SRAMs for storing input, weight, output, and ping-pang operations for hiding the latency of data moving between DRAM and SRAM. In order to reduce the power consumption of memory access, the system supports data broadcast operations from DARM to SRAM, which are controlled by DMA.

Because of the area and cost constraints of embedded devices, the SRAM size is limited by silicon cost. The whole data of CNN are stored in DRAM and only part of the data to be immediately calculated are stored in SRAM. It is tricky to choose the SRAM area size. If it is large enough, all the data of one convolutional layer are stored in the SRAM. There is no data wait caused by memory access, so the performance is completely dependent on the number of MACs. If it is too small, DLP needs to frequently access DRAM. The computing unit cannot be fully loaded, and the performance basically depends on the data access latency. Therefore, we need to balance performance and cost in choosing the appropriate SRAM.

Considering some real-time embedded applications, we used the ping-pang memory structure to minimize data access latency. As shown in Figure 7, while DLPs are running, the output tensor of the previous group of subtasks (blue blocks in the figure) can be stored in DRAM, and the weight and input tensor of the next group of subtasks (purple blocks in the figure) can be loaded to SRAM. To simplify the analysis, we assume that adjacent subtasks have the same amount of computation and data. If the total access latency is less than the computation latency, DLPs cannot perceive the data transition. This can be quantified as follows:(1)$$\begin{array}{c} \frac{BW\cdot U_{bus}}{N_{DLP}\cdot DA}\geq \frac{N_{MAC}\cdot Freq\cdot U_{PE}}{N_{op}} \end{array}$$
where BW and $U_{bus}$ are the bandwidth and utilization of the bus. $N_{DLP}$, $N_{MAC}$, and $U_{PE}$ are the number of DLP and MAC units, and the utilization of DLP, which represents the computing power of the system. DA and $N_{op}$ are the amounts of access and operation of the subtask. We assume the off-chip memory is DDR4, and its bandwidth is 42.6 GBPS (Gbyte Per Second). Therefore, we define compute density and its requirements as
(2)$$\begin{array}{c} \mathrm{Compute}\;\mathrm{desity}=\frac{N_{op}}{DA}\geq \frac{N_{MAC}\cdot Freq\cdot U_{PE}\cdot N_{DLP}}{BW\cdot U_{bus}}=\frac{\left(32\times 16\times 0.2\times 16\right)GMPS}{\left(42.6\mathrm{GB}/s\right)GBPS}=38 \end{array}$$

We traversed many possible subtasks to calculate their storage and compute density. Taking K = 3 and S = 1 as an example, the relationship between the storage and compute density is shown in the Figure 8. Each point represents a subtask [Th/Tl/Tn/Tm]. It can be seen that the compute density and memory size are positively correlated within a certain range of values. Therefore, when the compute density of the subtask is exactly 38, its storage requirement is the potential minimum on-chip storage. After the typical parameters are filtered, the memory sizes with the ping-pang operation of input, weight, and output tensor are 10 K, 14 K, and 8 K.

## 5. Scheduling Framework

Based on the hardware design proposed in the previous chapters, we analyzed the scheduling framework through software-hardware collaboration in this chapter.

### 5.1. The Main Challenges of CNN Acceleration

In the embedded inference acceleration of CNN, a great challenge is to trade off flexibility, performance, power efficiency, and area. For flexibility, by designing the application specific instruction set based on the interface operator of caffe2, our system naturally supports most CNN applications. For the area of this system, we choose the appropriate storage to balance performance and area cost as covered in the previous section.

Performance depends on the number of MACs, utilization, and memory access latency. In the previous section, we proposed a data-parallel scheme and DLP with 512 8-bit fixed-point MAC units, which illustrates that computing power is constant. Therefore, the opportunity to optimize performance involves improving utilization and reducing memory access latency.

Since the computing power is increased by expanding multiple processors, the utilization includes DLP’s PE utilization and DLP-level utilization. Because the execution of DLPs starts at the same time and the next subtask is not issued until all DLPs have completed execution, waiting between DLPs will reduce DLP-level utilization. Subtasks should be issued as evenly as possible on multiple DLPs to obtain higher utilization. Memory access latency is the main bottleneck for performance [21]. The latency of data access, especially DRAM, should be minimized to achieve high performance.

Energy consumption consists of energy consumed by data access and computation. For a certain CNN, the power of the computing operations is constant. The energy consumption of memory access is the sum of SRAM and DRAM access energy consumption. Because 32-bit DRAM access uses about a hundred times more energy than that of SRAM access [22], the main challenge in reducing power consumption is to reduce DRAM access.

### 5.2. The Mapping of CNN

There are two levels of mapping corresponding to three types of storage units, as shown in Figure 9a. The first level is task scheduling, also known as scheduling framework, which converts the convolutional layer stored in DRAM to subtasks stored in SRAM of DLPs. The data of one convolutional layer (H × L × N and R × C × M) is tiled into several subtasks (Th × Tl × Tn and Tr × Tc × Tm) that satisfy memory and other constraints. The pseudocode of the task scheduling is shown in Figure 9b. The for loop is the pseudocode of one convolution layer. The for loop unrolling and sequencing are assigned as subtasks to each DLP. The different unrolling and order of for loops will affect power consumption and performance. The second layer is the SIMD mapping, which transforms subtasks into instructions executed in DLPs. As shown in the pseudocode, Tn/Tm continues to expand to Tn_p/Tm_p for parallel computing. SIMD mapping mainly affects the parallelism of the datapath of DLP, which is not the focus of this article. Therefore, we mainly introduce task scheduling in the next section.

In the scheduling framework, we define the optimization objective to optimize utilization and power consumption. The computational power consumption is constant for certain applications, and DRAM accesses generate a major portion of the remaining power consumption. Therefore, the objective is defined as
(3)$$objective=\frac{U_{PE}}{E_{DRAM}}$$
where $E_{DRAM}$ is the energy consumption for DRAM access. $U_{PE}$ is the system-level utilization.
(4)$$\begin{array}{c} U_{PE}=\frac{\sum U_{MAC}}{PE}\;U_{MAC}=\;\mathrm{mod}(\frac{Tn\cdot Tm}{\mathrm{parallelism}}) \end{array}$$
where $U_{MAC}$ reflects the MAC unit utilization of one DLP. The DRAM access can be quantified as follows:(5)$${DA}_{DRAM}={DA}_{tile}+{DA}_{extra}-{DA}_{reduced}$$

$DA_{tile}$ is the data access of each subtask. $DA_{extra}$ refers to the additional data access generated by the partial output being accumulated as the final output. $DA_{reduced}$ is the memory access reduced by optimization.

If there is a large enough on-chip memory, the data of the neural network only need to be loaded once, achieving minimal memory access. Considering the area overhead of mobile applications, this is not realistic. Therefore, the layer of the neural network is tiled into several subtasks to satisfy practical memory constraints, while causing redundant data access.

However, these subtasks have data dependencies, and the way subtasks are mapped provides opportunities to reuse and broadcast data. Data reuse is data sharing between multiple subtasks. The same data of chronological subtasks of one DLP can remain stored in SRAM, which can reduce memory access by not loading these data again from DRAM. The data broadcast is bus sharing between multiple DLPs. The subtask data of different DLPs can be stored in the dedicated SRAM at one time when these data are the same. Hence, data broadcast can reduce memory access by avoiding the bus to repeatedly load the same data.

The reduced data access resulting from data reuse and data broadcast can be expressed as
(6)$$\begin{array}{c} DA_{reduced}={DA}_{reused}+{DA}_{broad}\\ DA_{broad}=pe_{broad}\cdot t_{broad}\cdot T_{broad}\\ DA_{reuse}=\left\{\begin{array}{c} pe_{\mathrm{reuse}}\cdot t_{reuse}\cdot T_{\mathrm{reuse}},reuse\;others\\ 2pe_{\mathrm{reuse}}\cdot t_{reuse}\cdot T_{\mathrm{reuse}},reuse\;output \end{array}\right. \end{array}$$
where $pe_{reuse}$ and $pe_{broad}$ are the number of DLPs whose data can be reused and broadcasted. $T_{reuse}$ and $T_{broad}$ refer to the data access of each subtask reduced by data reuse and broadcast. $t_{reuse}$ and $t_{broad}$ are the times of reuse and broadcast. If the output is reused, $DA_{extra}$ can also be reduced. Hence, its memory access reduction is twice that of others.

### 5.3. Reuse and Broadcast Pattern

In this section, several different data reuse and broadcast patterns are proposed, as shown in the Figure 10. The *X*-axes of (a,c,e,g) refer to the execution time of a subtask, and Y represents each slave processor. The blue blocks, such as K1, illustrate subtasks executed by each slave core. Figure 10b,d,f,h refer to the pseudocode of different reuse and broadcast patterns. The dashed lines are the pseudocode of one subtask, corresponding to blue blocks, such as K1.

(1) Output Reuse/Input Broadcast: As shown in Figure 10a, we propose a pattern called output reuse and input broadcast. The DMA needs to load input and weight data from DRAM to SRAM before accumulation. Following the computing of the subtasks, the DMA sends partial output data to DRAM. Finally, all partial results are accumulated to obtain the output.

The OR/IB can be transformed as the pseudocode in Figure 10b, and the whole pseudocode represents all subtasks of one convolution layer. The dashed box is one such subtask that is executed in the DLP. The loops C, R, M are the outer loops of Loop N, meaning the output is reused. A loop is expanded into multiple loops at the Tm level, and presents input that is broadcasted. The reduced data access of OR/IB is expressed as follows
(7)$$\begin{array}{c} pe_{\mathrm{reuse}}=pe_{\mathrm{broad}}\cdot \left\lfloor \frac{H}{Tl}\right\rfloor \cdot \left\lfloor \frac{L}{Tl}\right\rfloor \times \frac{16}{15}\;t_{reuse}=\left\lfloor \frac{N}{Tn}\right\rfloor -1\\ pe_{\mathrm{broad}}=\left\lfloor \frac{\left\lfloor \frac{M}{Tm}\right\rfloor }{16}\right\rfloor \times \left(16-1\right)\;\;\;\;\;\;t_{broad}=\left\lfloor \frac{N}{Tn}\right\rfloor \cdot \left\lfloor \frac{H}{Th}\right\rfloor \cdot \left\lfloor \frac{L}{Tl}\right\rfloor \\ T_{reuse}=Th\cdot Tl\cdot Tm\;\;\;\;\;\;T_{broad}=Th_{in}\cdot Tl_{in}\cdot Tn \end{array}$$

In order to simplify the design of the DMA, pe reuse and pe broad of the subtask are the integer times of the DLPs, otherwise, it will cause some DLP idleness and reduce the overall system utilization.

(2) Output Reuse/Weight Broadcast: Similar to OR/IB, the output reuse and weight broadcast pattern is presented, as shown in Figure 10c. The Ci layer and Hi layer of the convolutional layer are tiled together. The same weight of each subtask can be broadcasted from DRAM to SRAM. The pseudocode of OR/IB is shown as Figure 10d. Loop H and L are expanded to broadcast weight. Loop M is the outer loop of Loop N to reuse input. Because some convolution layers are difficult to reuse and broadcast data at the same time, we also propose a mode that only reuses the output.
(8)$$\begin{array}{c} pe_{\mathrm{reuse}}=\left\{\begin{array}{c} \left\lfloor \frac{\left\lfloor \frac{H}{Th}\right\rfloor \cdot \left\lfloor \frac{L}{Tl}\right\rfloor \cdot \left\lfloor \frac{M}{Tm}\right\rfloor }{16}\right\rfloor \times 16,OR/NB\\ pe_{\mathrm{broad}}\cdot \left\lfloor \frac{M}{Tm}\right\rfloor \times \frac{16}{15},OR/WB \end{array}\right.\;t_{reuse}=\left\lfloor \frac{N}{Tn}\right\rfloor -1\\ pe_{\mathrm{broad}}=\left\{\begin{array}{c} 0,OR/NB\\ \left\lfloor \frac{\left\lfloor \frac{H}{Th}\right\rfloor \cdot \left\lfloor \frac{L}{Tl}\right\rfloor }{16}\right\rfloor \times \left(16-1\right),OR/WB \end{array}\right.\;t_{broad}=\left\lfloor \frac{M}{Tm}\right\rfloor \times \left\lfloor \frac{N}{Tn}\right\rfloor \\ T_{reuse}=Th\cdot Tl\cdot Tm\;\;\;\;\;\;\;\;\;\;\;T_{broad}=K^{2}\cdot Tm\cdot Tn \end{array}$$

(3) Input Reuse/Weight Broadcast: In order to reduce the memory access of the input map and weight, we introduce the input reuse and weight broadcast pattern. The IR/WB can be transformed into a pseudocode shown in Figure 10e,f. Compared to Loop H, L, and M, Loop N is the innermost loop to reuse the input. Meanwhile, we also consider the reuse-only pattern.
(9)$$\begin{array}{c} pe_{\mathrm{reuse}}=\left\{\begin{array}{c} \left\lfloor \frac{\left\lfloor \frac{H}{Th}\right\rfloor \times \left\lfloor \frac{L}{Tl}\right\rfloor \times \left\lfloor \frac{N}{Tn}\right\rfloor }{16}\right\rfloor \times 16,IR/NB\\ pe_{\mathrm{broad}}\cdot \left\lfloor \frac{N}{Tn}\right\rfloor \times \frac{16}{15},IR/WB \end{array}\right.\;\;t_{reuse}=\left\lfloor \frac{M}{Tm}\right\rfloor -1\\ pe_{\mathrm{broad}}=\left\{\begin{array}{c} 0,IR/NB\\ \left\lfloor \frac{\left\lfloor \frac{H}{Th}\right\rfloor \times \left\lfloor \frac{L}{Tl}\right\rfloor }{16}\right\rfloor \times \left(16-1\right),IR/WB \end{array}\right.\;\;\;t_{broad}=\left\lfloor \frac{M}{Tm}\right\rfloor \times \left\lfloor \frac{N}{Tn}\right\rfloor \\ T_{reuse}=Th_{in}\cdot Tl_{in}\cdot Tn\;\;\;\;\;\;\;\;\;\;\;\;T_{broad}=K^{2}\times Tm\times Tn \end{array}$$

(4) Weight Reuse/Input Broadcast: We present the weight reuse and input broadcast pattern as shown in Figure 10g; the pseudocode is shown in Figure 10h. A loop is split at the M level and assigned to each DLP to the broadcast input. Loop H and L are the inner loops of Loop N to reuse weights.
(10)$$\begin{array}{c} pe_{\mathrm{reuse}}=\left\{\begin{array}{c} \left\lfloor \frac{\left\lfloor \frac{N}{Tn}\right\rfloor \cdot \left\lfloor \frac{M}{Tm}\right\rfloor }{16}\right\rfloor \times 16,WR/NB\\ pe_{\mathrm{broad}}\cdot \left\lfloor \frac{N}{Tn}\right\rfloor \times \frac{16}{15},WR/IB \end{array}\right.\;\;\;t_{reuse}=\left\lfloor \frac{H}{Th}\right\rfloor \cdot \left\lfloor \frac{L}{Tl}\right\rfloor -1\\ pe_{\mathrm{broad}}=\left\{\begin{array}{c} 0,WR/NB\\ \left\lfloor \frac{\left\lfloor \frac{M}{Tm}\right\rfloor }{16}\right\rfloor \times \left(16-1\right),WR/IB \end{array}\right.\;t_{broad}=\left\lfloor \frac{H}{Th}\right\rfloor \cdot \left\lfloor \frac{L}{Tl}\right\rfloor \cdot \left\lfloor \frac{N}{Tn}\right\rfloor \\ T_{reuse}=K^{2}\cdot Tm\cdot Tn\;\;\;\;\;\;\;\;\;T_{broad}=Th_{in}\cdot Tl_{in}\cdot Tn \end{array}$$

### 5.4. Workflow

The workflow of the scheduling framework is shown in Figure 11. In this paper, we have multiple optimization objectives and, therefore, a multi-objective optimization problem. However, our performance is only a function of utilization and energy consumption, without considering memory access latency. To simplify the analysis, we transform the multi-objective problem into a single-objective optimization problem.

Since access delay is the performance bottleneck, it is prioritized over power consumption and utilization during optimization. In the section covering extension of the DLP compute capability, we proposed the compute density. Under the computing power and bandwidth of this paper, when the compute density of the subtask is greater than 38, there is no memory access delay due to hiding memory access delay in the calculation. In the previous analysis, the compute density did not take into account the bandwidth saved by data reuse and broadcasting. Therefore, the compute density is reduced to obtain more possible subtasks for optimization. These subtasks can also completely hide the memory access latency. The subtask ([Tn, Tl, Tn, Tm]) with the smallest access latency is selected as the selected pool, and the goal is then optimized based on the selected pool. In addition, the subtasks of the selected pool need to satisfy two constraints. The SIMD is parallel in the Tn and Tm dimensions. Hence, Tn × Tm should be a multiple of the number of MAC units such that the MAC utilization is 100%. The amount of data in the subtasks also needs to be smaller than the actual memory size.

For each layer, our framework acquires the final task scheduling scheme through data partition based on the selected pool, as shown in Figure 11. This scheduling scheme includes subtasks and data reuse and broadcast patterns. After tiling by [Th, Tl, Tn, Tm] from the selected pool, there may be epilogues in the H, L, N, and M dimensions. If the epilogue does not meet the memory size limits, it will be iteratively split until the amount of data of the epilogue is less than the memory capacity.

## 6. Experiment

In the previous section, we implemented the acceleration system through the co-design of software and hardware, including the Soc-level extension scheme, DLP, and scheduling framework. DLP is synthesized based on TSMC 65 nm LP technology using the Synopsys design compiler, until a machine clock speed of 200 M is reached. The compilation of RTL code in Verilog and functional simulation are conducted on Synopsys VCS, and the core power consumption is estimated by PrimeTime PX. Note that the Soc-level scheme is an experimental model without synthesizing. In this model, the off-chip memory is assumed to be DDR4, and its bandwidth is 42.6 GBPS (Gbyte Per Second).

We chose some typical convolutional neural networks, such as AlexNet, VGG, and GoogleNet for experiments. Since the time for computation of the convolutional layers occupies most of the total computation time [23], we mainly analyzed the convolutional layer of these CNN models. The Conv and FC layers are translated to subtasks that DLP can perform using the scheduling framework. These subtasks are manually translated into assembly code, and the assembly code is translated into a binary file using an assembler for our dedicated ISA. DLP is configured to run binary files, which are custom instructions corresponding to subtasks. We evaluate the whole network running on the acceleration system based on these subtasks.

### 6.1. Characteristics

The basic characteristics of the DLP and Soc-level scheme are shown in Table 7. The DLP is an AISP processor for the neural network proposed in this paper with 32 MAC units in SIMD parallel and 32 KB SRAM. The Soc-level scheme system includes 16 DLPs with 512 MACs and 512 KB SoC SRAM. The data precision of the MAC unit is 8-bit fixed point. If the PE utilization is 100%, the maximum performances of the DLP and Soc-level schemes are 12.8GOPS and 204.8GOPs. The average performance and computing power consumption of DLP are 12.25GOPS and 25.75 mW, measured on DNA’s netlist with the subtasks’ simulation wave files. The average result of the Soc-level scheme system is measured based on DLP, the experimental model, and the scheduling scheme of benchmarks.

### 6.2. Flexibility

Unlike other designs based on several concrete CNN models, our application-specific ISA is designed based on the inference and basic operators of caffe2. Therefore, we can support all networks using these operators. It makes perfect sense that algorithm researchers focus more attention on their innovation points rather than waste the majority of the time on adapting the accelerators.

In addition, due to our independent hardware design of memory and DLP, the computing power can be easily extended at the Soc-level for different applications. Our scheduling framework is also flexible enough to support these.

### 6.3. Results of Scheduling Framework

Each Conv layer of AlexNet is mapped to DLP using the scheduling framework. The results of the scheduling framework in the AlexNet Conv layer include subtasks ([Th, Tl, Tn, Tm]), reuse, and a broadcast pattern as shown in Table 8. For the Conv1 layer, the IR/WB pattern is used to reduce memory access. This is because H and L are large enough to broadcast weight and appropriate M to reuse the input tensor, while N is too small to optimize memory access for other patterns. The Conv2 layer uses a WR/IB pattern. The remaining Conv layers of AlexNet use an OR/IB pattern. Without OR, the partial sums of subtasks need to be stored in and loaded from DRAM to compute the final output before finishing the computation. Because OR can reduce the memory access of the output tensor twofold, the outputs of the Conv3-5 layers are reused to optimize power consumption. Note that the power consumption, utilization, and performance on AlexNet, VGGNet, and GoogleNet are analyzed based on this type of scheduling scheme.

Figure 12 shows the simulation waveform of an AlexNet Conv3 subtask. The red cursor 1 indicates the start of the calculation and the configuration of subtask parameters, such as base address and fixed-point number format. The red cursors 2 and 3 indicate the start and end of the calculation. When mac_vld is 1, the data is calculated in the MAC units of the SIMD lane. Considering the design complexity, the repeat instruction supports up to three layers of loops. There are traditional conditional jump instructions in the program to support the convolution layer, which will cause flush. Therefore, during the operation, mac_vld became 0 at some point. The operation of DLP accessing data from SRAM is not shown in the simulation waveform. This is because we insert the SRAM reading and writing stages into the DLP pipeline, by which the reading/writing process of data can be hidden into the computing process. Although the pipeline is extended in this way, it is affordable since the latency caused by a long pipeline is ignorable compared with the latency caused by a separate load/store instruction.

### 6.4. Energy Consumption and DRAM Access

Energy consumption consists of energy consumed by data access and computation. We measure the energy consumption of the network based on the synthesizing result of DLP and the experimental model of the Soc-level scheme. The estimated power consumption results can be quantified as
(11)$$E_{total}=E_{DRAM}\cdot {DA}_{DRAM}+P_{DLP}\cdot Time\cdot N_{DLP}$$
where $E_{total}$ and $E_{DRAM}$ are the energy consumption of the system and per data access of DRAM. $E_{DRAM}$ is estimated as 50 pJ/bit [24]. $P_{DLP}$ is the computing power of the core. $P_{DLP}$ and time are measured based on the DLP’s simulation wave files of the convolutional layer subtask with PrimeTime PX and Synopsys VCS. $N_{DLP}$ is the number of DLP in the Soc-level scheme, which is 16 in this paper. ${DA}_{DRAM}$ is DRAM access and is counted with Equation (Equation 5) based on the task scheduling scheme of the benchmarks.

There are three dimensions to analyze DRAM access, all Conv layers of AlexNet, one Conv layer of AlexNet, and different CNN models.

First, according to the task scheduling scheme of each layer, the DRAM access is shown in Figure 13a, where R/B illustrates that there is data reuse and broadcast. NR/NB has the opposite meaning. It can be seen that R/B can effectively reduce memory access. For the Conv1 layer, the memory access of NB is 1.24 times that of B. As for the Conv2-4 layers, R/B reduced the DRAM access by 60.9%, 28%, 37.5%, and 37.5%, respectively, compared with NB.

Second, the breakdown analysis on the Conv2 layer of AlexNet for different data reuse and broadcast patterns is shown in Figure 13b. It can be seen that the memory access of WR/IB is the smallest. The memory access of OR/IB and WR/IB are 21% and 46% lower than that of OR/NB and WR/NB, meaning the data broadcast is effective in reducing memory access.

Lastly, to demonstrate the superiority of our layer-based task scheduling over network-based task scheduling, we analyzed the memory access of several networks, such as AlexNet, VGG, and GoogleNet. There are two modes, hybrid (corresponding to layer-based) and single (corresponding to network-based). The hybrid mode is that each layer of neural networks can use different data reuse and broadcast patterns. The single mode is that all layers of neural networks only use the same data reuse pattern, including IR, OR, and WR. The easiest way is single mode, because it only needs to configure the data reuse and broadcast pattern once at the beginning of the network, while the hybrid mode requires additional configuration at each layer.

However, the hybrid mode can minimize memory access. As shown in Figure 14, the hybrid mode achieves a memory access reduction of 5.5 times greater than for IR, 9.7 times greater than OR, and 6.8 times greater than WR on VGG16Net. The hybrid pattern also optimizes memory access well in the AlexNet and GoogleNet, as shown in Figure 14.

### 6.5. Performance

There are many factors affecting performance, such as memory access latency, the number of MAC units, control latency, and utilization. However, for a certain system, the number of computational units is constant. Other factors are analyzed as follows.

(1) Control latency

The repeat instruction in our ISA can significantly reduce control latency. In Figure 9b, [Tn, Tm] is tiled into [Tn_p, Tn_b, Tm_p, Tm_b] for parallel computation. Loop Tn_p and Tm_p can be implemented by one SIMD instruction. Loop Tn_b, Tm_b, and K can be simplified to a repeat instruction for acceleration. In this way, the repeat instruction reduces the control latency by 97%.

(2) Utilization

The PE utilization ratio on the benchmarks is defined as 1 = 100%; the hardware is fully used/activated when running the algorithm. The PE utilization of our design on benchmarks is shown in Figure 15. The datapath of DLP is 1-D parallelization in N and M dimensions. Therefore, if the N dimension times the M dimension of a subtask is a multiple of the parallelism, the PE utilization is 1. In order to pursue higher utilization, the subtasks whose Tn × Tm is not a multiple of 32 will be discarded in the scheduling framework. Note that the utilization of the first layer of this network is 94%. The N of the AlexNet Conv1 is only 3, which makes it difficult to efficiently map to DLP. Finally, our design achieves an average utilization of 98.75%, 99.8%, and 98.7% in AlexNet, VGGNet, and GoogleNet, respectively.

(3) Memory access latency

The compute density is proposed to quantify the reduction in memory access latency in Section 4. In our experimental model, when the computing density of the subtask is greater than 38, the latency of total DRAM access is less than the computation latency of DLPs. With ping-pong memory, DLPs cannot perceive the data transition. The compute density of the convolutional layers of AlexNet are 37, 45, 39, 39, and 39. Note that the compute density of Conv1 is 37. The computing density did not take into account the bandwidth saved by data multiplexing and broadcasting in the previous analysis. Therefore, our scheme can hide total memory access latency in AlexNet Conv1.

### 6.6. Design Comparison

As shown in Table 9, we compare our design with several state-of-the-art implementations, such as GPU and Eyeriss, [25,26] on benchmarks.

GTX 1080 Ti is selected as the GPU for evaluation. Its clock frequency and memory are 1582 MHz and 11 GB. It has high performance, but its high power consumption means it is not appropriate for embedded applications. The examples in [5,25,26] are accelerators for deep learning, working at 250, 1500, and 400 MHz, respectively. The c and s in the table represent chip-only power and system-level power considering off-chip memory.

The architecture in [25] and our work are the most flexible to implement neural network applications because their architecture is ASIP, whose ISA is customized for deep learning. The [25] architecture is designed for deep learning training and inference However, it consists of a 2-D compute array, which leads to complex design and control separation. DLP is designed with a 1-D MAC array, which has simpler control logic and is more conducive to accelerating element-wise type operations. The DLP has only 32 MAC units, which can be expanded more flexibly to support different applications.

Our performance reached 196 GOPS considering DRAM access latency, unlike most designs that do not consider it. The performance is almost at its limit because of our optimization in utilization and memory latency. The hardware design of DLP is 1-D parallelization, which is flexible enough to issue subtasks for the highest utilization. Therefore, the utilization of our design reaches 99%, which is 30%, 6%, and 4% higher than [5,25,26], respectively. Other designs, such as [25], use the large on-chip memory to minimize the access delay between DRAM and on-chip memory. However, our work can reduce all DRAM access latency in convolution with smaller on-chip memory, through ping-pang memory, as well as the scheduling framework. We define memory efficiency as the number of MAC units divided by the on-chip storage. Our design achieves a memory efficiency of 1, which is 1.1, 4.2, 1.5 times that of [5,25,26], respectively. Since the scheduling framework is proposed to reduce memory access through data reuse and broadcast, our design achieves 813 mW on benchmarks. Although our energy efficiency ratio is slightly worse than [26], we consider the DRAM access latency and have higher efficiency per MAC/memory. Finally, while achieving high flexibility, the power efficiency of our design for our benchmarks is 241 GOPS/W.

## 7. Conclusions

In this paper, we proposed an application-specific ISA based on the caffe2 inference operator, which can flexibly accelerate the inference of most neural networks. The corresponding deep learning processor is synthesized in TSMC 65 nm LP technology. The DLP works at 200 MHz with 32 8-bit fixed-point MAC units and 32K SRAM. Because the compute capability of one DLP is limited to DNN accelerations, we proposed the Soc-level extension scheme with the scheduling framework to optimize the performance and energy consumption. The bottleneck of power consumption and performance is mainly due to the power consumption and delay generated by memory access. To improve performance, the VLIW and ping-pang memory architecture were used to eliminate all data access time costs for the convolution layer. To reduce power, the data access was reduced by reusing and broadcasting data. Finally, our design achieved 196 GOPS at 200 MHz and the power efficiency was 241 GOPS/W on the VGG16Net and AlexNet. Our future work includes the complete implementation of the Soc expansion scheme, the final chip layout and tape-out of the prototype design, and trade-off between the bus bandwidth and parallelism to reduce memory access latency for different applications.

## Figures and Tables

**Figure 1 sensors-22-03841-f001:**
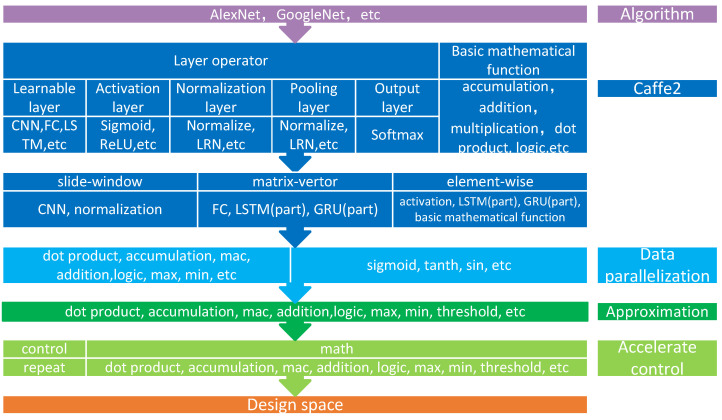
The design space extracted from caffe2.

**Figure 2 sensors-22-03841-f002:**
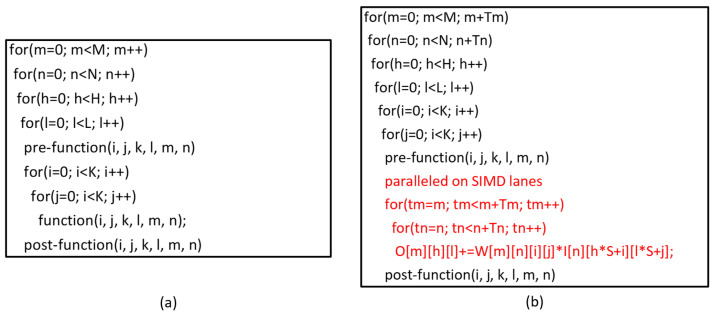
Pseudocode for (**a**) fused algorithm and (**b**) paralleled convolution.

**Figure 3 sensors-22-03841-f003:**
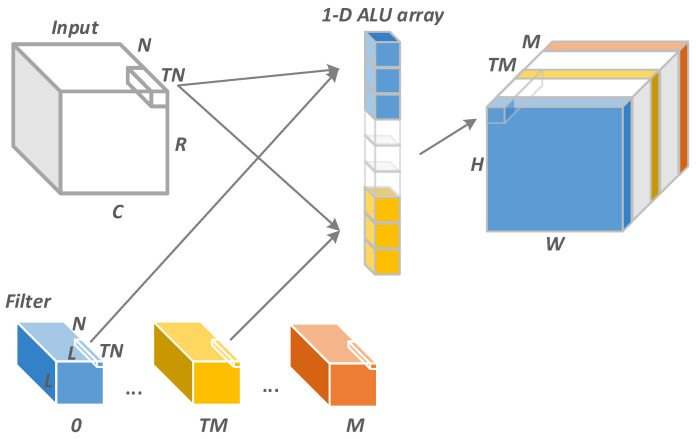
Convolution parallelization along input channel and filter batch dimension on a 1-D MAC array.

**Figure 4 sensors-22-03841-f004:**
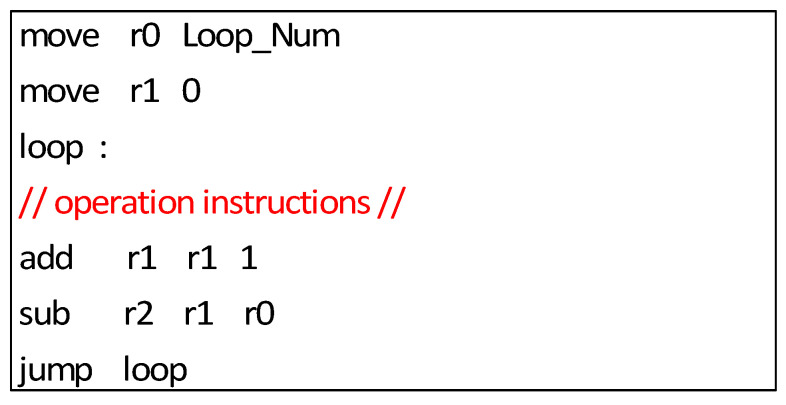
Traditional loop assembly code.

**Figure 5 sensors-22-03841-f005:**
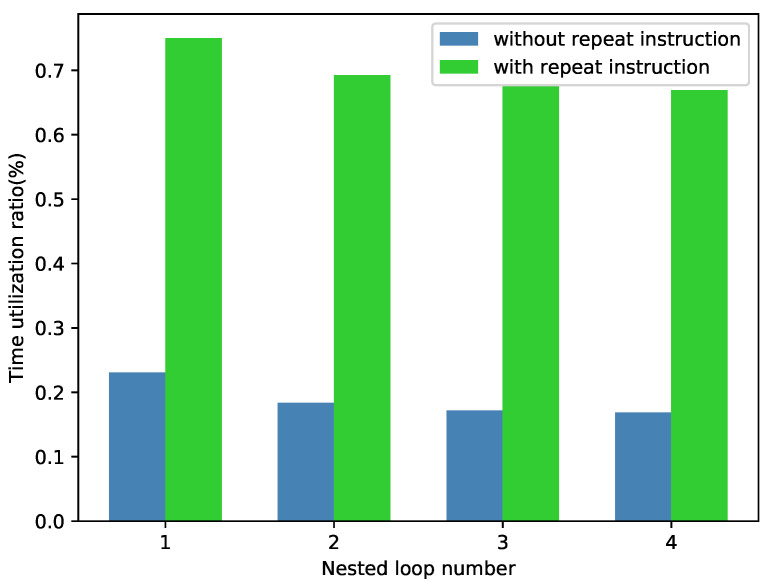
The time utilization ratio comparison between the code with/without repeat instruction.

**Figure 6 sensors-22-03841-f006:**
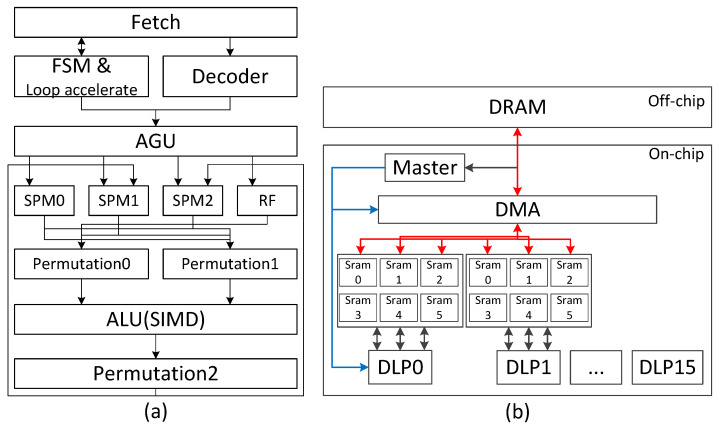
The hardware design. (**a**) PE micro-architecture. (**b**) Soc-level extension scheme.

**Figure 7 sensors-22-03841-f007:**
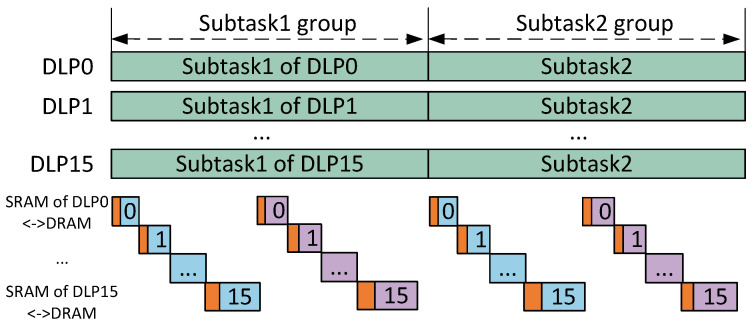
The relationship between the latency of access and the computing time.

**Figure 8 sensors-22-03841-f008:**
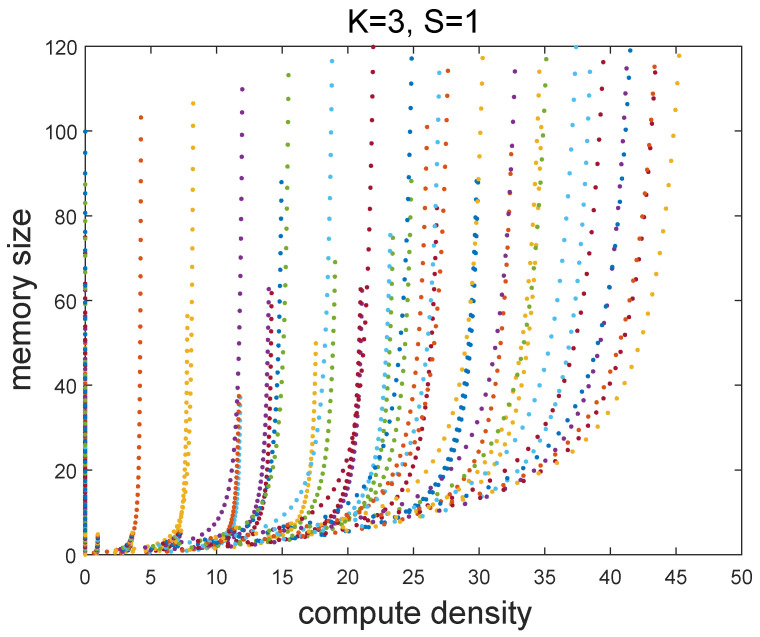
The relationship between memory and compute density.

**Figure 9 sensors-22-03841-f009:**
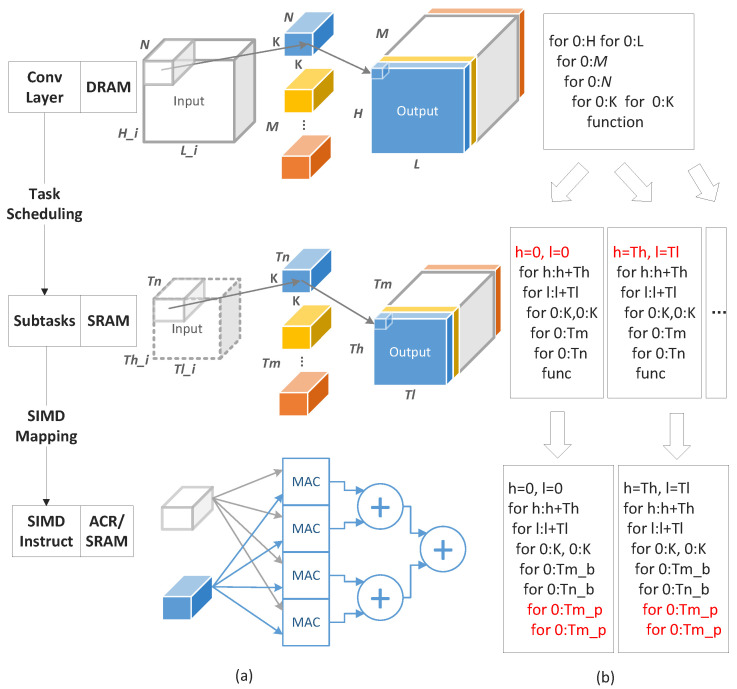
Mapping of convolution layer. (**a**) Mapping from convolution layer to DLPs. (**b**) Corresponding pseudocode.

**Figure 10 sensors-22-03841-f010:**
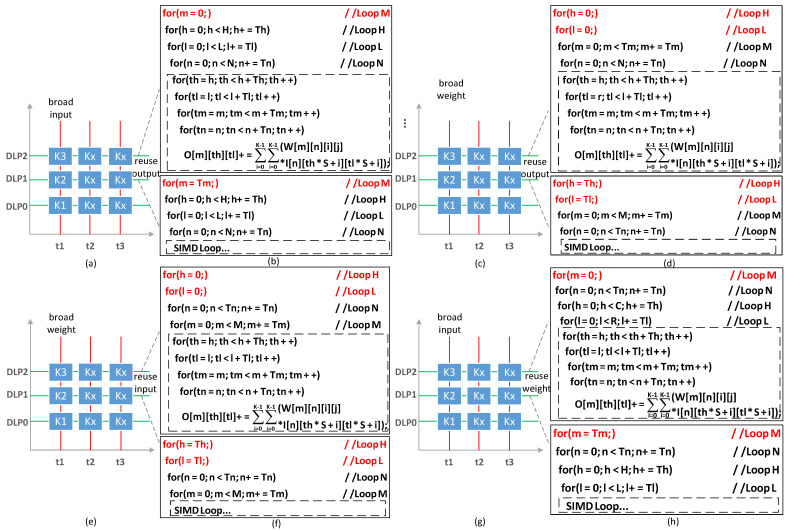
Data reuse and broadcast pattern: OR/IB, OR/WB, IR/WB, WR/IB (**a**,**c**,**e**,**g**) refer to their execution. (**b**,**d**,**f**,**h**) refer to their pseudocode.

**Figure 11 sensors-22-03841-f011:**
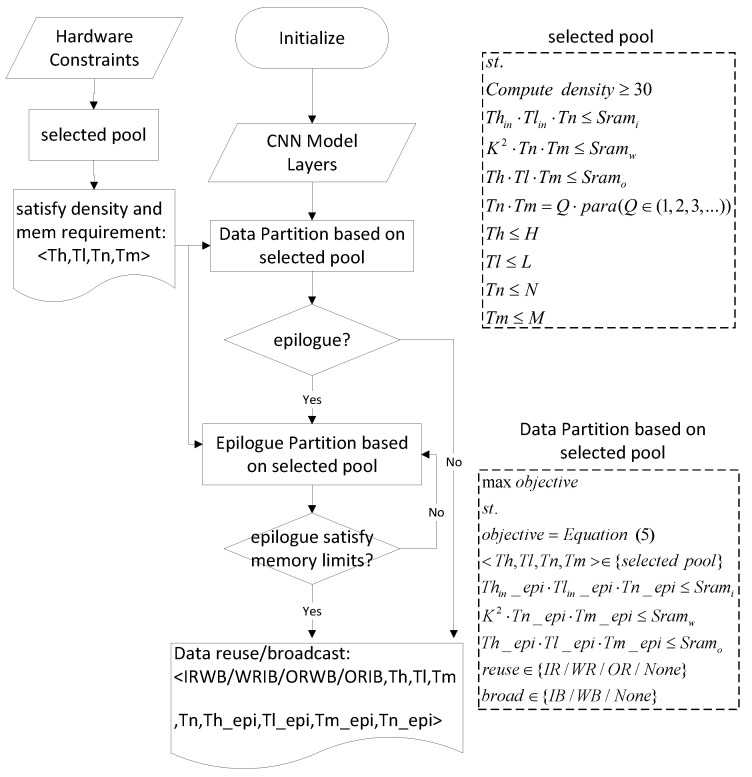
The workflow of the schedule framework.

**Figure 12 sensors-22-03841-f012:**
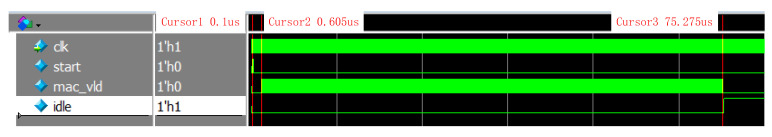
Simulation waveform of a subtask.

**Figure 13 sensors-22-03841-f013:**
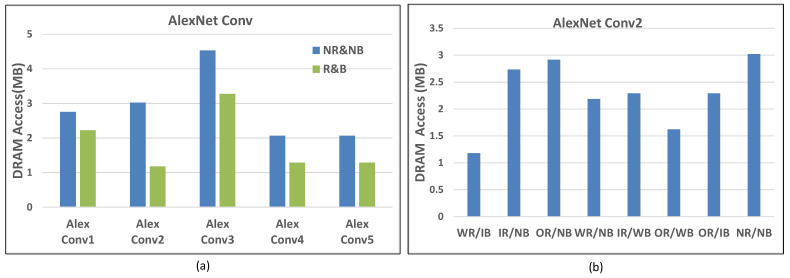
The DRAM access of convolution layers of AlexNet. (**a**) AlexNet Conv1-5. (**b**) AlexNet Conv2.

**Figure 14 sensors-22-03841-f014:**
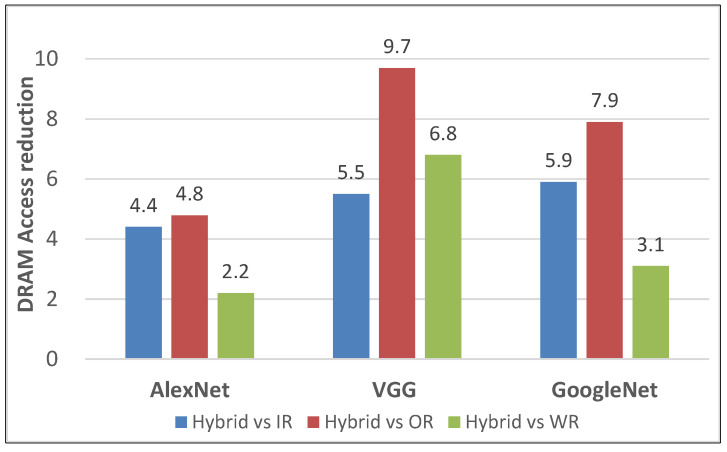
The memory access reduction: hybrid mode over single mode (IR/OR/WR).

**Figure 15 sensors-22-03841-f015:**
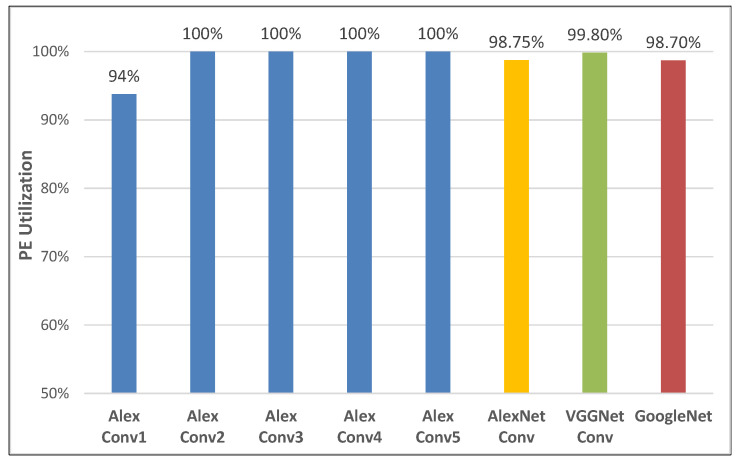
PE utilization.

**Table 1 sensors-22-03841-t001:** The framework components for different kernel types.

	Loop6	Loop5	Loop4	Loop3	Loop2	Loop1	Function1	Function2
window-sweep	*√*	*√*	*√*	*√*	*√*	*√*	*√*	*√*
martix-vector mul					*√*	*√*	*√*	*√*
element-wise				*√*	*√*	*√*	*√*	*√*

**Table 2 sensors-22-03841-t002:** Approximation functions.

Function	Approximation Method	Approximation Expression
sigmoid	segment	$\mathrm{Hsigm}\left(x\right)=\left\{\begin{array}{c} +1,if\;x>4\\ \frac{x}{4}+0.5,otherwise\\ 0,if\;x\leq -1 \end{array}\right.$
tanh	segment	$\mathrm{Ptanh}\left(x\right)=\left\{\begin{array}{c} +1,if\;x>1\\ \frac{x}{4}+\frac{3}{8},if\;0.5<x\leq 2.5\\ x,otherwise\\ \frac{x}{4}-\frac{3}{8},if\;-2.5<x\leq -0.5\\ -1,if\;x\leq -2.5 \end{array}\right.$
sin	Taylor	$\sin\left(x\right)=x-\frac{x^{3}}{6}$

**Table 3 sensors-22-03841-t003:** Overview of the instructions.

Operation		Notation	Type
compute	add	add	type1
	subtract	sub	type1
	multiply	mul	type1
	multiply and accumulate	mac	type1
	maximum	max	type1
	triangular accumulate	tacc	type2
logic	not	not	type2
	and	and	type1
	or	or	type1
shift	shift right	shr	type1
	shift left	shl	type1
control	no operation	nop	type4
	clear accumulator register	car	type4
	jump	jmp	type5
	repeat	repeat	type6
data transfer	copy	copy	type2
	load immediate operand	ldi	type3

**Table 4 sensors-22-03841-t004:** Instruction type.

Type	Expression					
	8 bit	4 bit	7 bit	13 bit	16 bit	16 bit
type1	opcode	cdt	opt	dst	src0	src1
type2	opcode	-	-	dst	src0	-
type3	opcode	-	-	dst	-	-
type4	opcode	-	-	-	-	-

**Table 5 sensors-22-03841-t005:** Option and condition type.

		Notation
option	shift	shiftr = 0–15/shiftl = 1–15
	round	rnd
	saturation	sat
condition	equal	eq
	not equal	neq

**Table 6 sensors-22-03841-t006:** Address type.

Operand		Segment			Notation
src0/src1	dst	10 bit	addr	reg	reg
				LVM addr by reg	[reg]
				imm16	imm16
				accumulator register	acr
		3 bit	length		i/ii/iii/iv/v/vi
	-	3 bit	pattern		w/d/f/e/h/v

**Table 7 sensors-22-03841-t007:** The characteristics of DLP and Soc-level schemes.

	DLP	Soc-Level Scheme
Technology	TSMC 65 nm LP	None
SRAM	32 KB	512 KB
Frequency	200 Mhz	200 Mhz
MAC unit	32	512
Peak Performance	12.8GOPS	204.8GOPS
Average Performance	12.25GOPS	196GOPS
Average Power	25.75 mW	813 mW
Precision	8-bit Fixed Point	8-bit Fixed Point

**Table 8 sensors-22-03841-t008:** The scheduling scheme of AlexNet Conv layers.

AlexNet	<Th, Tl, Tn, Tm>	<Reuse, Broad>	Compute Density
Conv1	<27, 79, 2, 16>	<IR, WB>	37
Conv2	<7, 13, 48, 4>	<WR, IB>	45
Conv3	<13, 13, 16, 12>	<OR, IB>	39
Conv4	<13, 13, 16, 12>	<OR, IB>	39
Conv5	<13, 13, 16, 12>	<OR, IB>	39

**Table 9 sensors-22-03841-t009:** Comparison with state-of-the-art designs.

	GTX 1080 Ti	ISSCC16 [5]	VLSI18 [25]	ISSCC22 [26]	This Work
Process	16 nm	65 nm	14 nm	65 nm	65 nm
Architecture	GPU	CNN	ASIP	CPU + CNN	ASIP
Benchmark	-	AlexNet	ResNet18ResNet50	VGG16ResNet18	VGG16AlexNet
Frequency	1582 MHz	250 MHz	1500 MHz	400 MHz	200 MHz
MAC Number	1582 Core	168	500	100	512
Bit Frequency	FP32	INT16	FP16/32	INT8	FXP8
SRAM	11 GB	181 KB	2048 KB	150 KB	512 KB
MAC/SRAM	-	0.93	0.24	0.67	1
PE Utilization	-	68.70%	92 98%	95%	99%
Efficiency (TOPS/W)	0.041	c: 0.166 s: 0.08	-	c: 0.66(CNN)	c: 0.475 s: 0.241

## Data Availability

Not applicable.

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
