# Peer review of "An ASIP for Neural Network Inference on Embedded Devices with 99% PE Utilization and 100% Memory Hidden under Low Silicon Cost"

_sensors, 2022, doi:10.3390/s22103841_

Round 1
Reviewer 1 Report
Authors conducted this research in the title of “An ASIP for Neural Network Inference on Embedded Devices with 99% PE utilization and 100% memory hidden under low silicon cost".
The paper’s subject could be interesting for readers of journal. Therefore, I recommend this paper for publication in this journal but before that, I have a few comments on the text that should be addressed before publication:
Comments:
- Abstract: In the Abstract section, authors should pay more attention to the main goals and questions that are supposed to be addressed in this article. It would be really helpful for readers of this paper because they first read abstracts to know if an article interests them or is related to a subject important to them. Instead of checking numerous written materials, readers depend on abstracts to quickly determine if an article is relevant to them or not.
- Line 5 in Abstract section: authors used this word “We” at the beginning of the sentence. The beginning of a sentence is a noticeable position that draws readers’ attention. Thus, using personal pronouns as the first one or two words of a sentence will draw unnecessary attention to them, so it is better to avoid starting a sentence with personal pronouns. For example, this sentence “We presented a new model in this work” could be rewritten as “a new model is presented in this work”.
- Figure 10: The title of this figure is too long. Authors could explain more after or before this figure. In other words, captions should be concise but comprehensive. They should describe the data shown, draw attention to important features contained within the figure.
- Figure 12: There is a table without any relevant title at the left side of Figure 12. Furthermore, this table should be moved to another place. The current position of the table is not appropriate.
- Table 7: This table could be dragged more to the right direction in order to put it in the middle alignment. It really looks better and more integrated with other tables in the article.
- Page 19, Conclusion: The Conclusion section is too short and concise. In the conclusion section authors should mention more words about their suggestions to future works. It really can be helpful for future studies and works related with title of this article. For example, addressing limitations of your research, your research will not be free from limitations and these may relate to formulation of research aim and objectives, application of data collection method, sample size, scope of discussions and analysis etc. You can propose future research suggestions that address the limitations of your study.
- Which software has been used in this work to export the charts and diagrams in this work? For instance, software like SigmaPlot or SmartDraw are used to export and depict charts. Mentioning used software would be helpful to future researches and studies in the field of this article.
- Page 14, Figure 10: Some used images in this figure (especially the right bottom images) are unclear and it is too hard to read them. The most important consideration for figures is simplicity. Choose images the viewer can grasp and interpret clearly and quickly. Consider size, resolution, color, and prominence of important features. Figures should be large enough and of sufficient resolution for the viewer to make out details without straining their eyes.
- Keywords: The word “ISA“ could be mentioned in this section because it has been used repeatedly in different parts of this paper and it seems a keyword. The purpose of keywords in a research paper is to help other researchers find your paper when they are conducting a search on the topic. Keywords define the field, subfield, topic, research issue, etc. that are covered by the article. Most electronic search engines, databases, or journal websites use keywords to decide whether and when to display your paper to interested readers. Keywords make your paper searchable and ensure that you get more citations.
- Since recently it has been proved that artificial intelligence (AI) and machine learning has a numerous applications in all of engineering fields, I highly recommend the authors to add some references in this manuscript in this regard. It would be useful for the readers of journal to get familiar with the application of AI in other engineering fields. I recommend the others to add all the following references, which are the newest references in this field
[1] Chenarlogh, V. A., Razzazi, F., & Mohammadyahya, N. (2019, December). A Multi-View Human Action Recognition System in Limited Data Case using Multi-Stream CNN. In 2019 5th Iranian Conference on Signal Processing and Intelligent Systems (ICSPIS) (pp. 1-11). IEEE.
[2] Roshani, M., et al. 2020. Application of GMDH neural network technique to improve measuring precision of a simplified photon attenuation based two-phase flowmeter. Flow Measurement and Instrumentation, 75, p.101804.
[3] Jafari Gukeh, M., Moitra, S., Ibrahim, A. N., Derrible, S., & Megaridis, C. M. (2021). Machine Learning Prediction of TiO2-Coating Wettability Tuned via UV Exposure. ACS Applied Materials & Interfaces, 13(38), 46171-46179..
[4] Azimirad, V., Ramezanlou, M. T., Sotubadi, S. V., & Janabi-Sharifi, F. (2021). A consecutive hybrid spiking-convolutional (CHSC) neural controller for sequential decision making in robots. Neurocomputing.
Reviewer 2 Report
In this paper, the authors proposed an application-specific ISA based on the caffe2 inference. The corresponding deep learning processor was implemented in TSMC 65-nm technology. The proposed SoC make use of a so-called scheduling framework to optimize the performance and energy consumption.
While this work is important but it requires major changes to elaborate the genuine contributions and the new knowledge it creates. In its present form, it is very hard to ascertain the actual contribution of the paper. Please see below my general and specific concerns that need addressing.
General comments:
- This article does not include all the necessary details. The contributions claimed by the authors need to be backed up by experimental, quantifiable, retrospective data.
- The authors highlighted that they focused on low power and low latency data accesses and achieved a satisfactory result. Could the authors quantify what they mean by “satisfied” results in the abstract of the paper? Satisfied is a relative term!
- There are typos all over the manuscript and require rigorous proofreading.
- All figures should be explicitly defined in the manuscript and highlighted. Few figures are missing in the text. For example, figures 3, and 4, are not referred to anywhere in the manuscript.
- This paper seems to be a system paper but the system information is missing. The authors chose to include very few details with no mention of the chip design, layout, characterization, benchmarking and so on.
- In the introduction section where the hardware accelerators are defined. Please include the below-mentioned most recent paper on deep learning-based hardware accelerators.
Ghani, A.; Aina, A.; See, C.H.; Yu, H.; Keates, S. Accelerated Diagnosis of Novel Coronavirus (COVID-19)—Computer Vision with Convolutional Neural Networks (CNNs). Electronics 2022, 11, 1148. https://doi.org/10.3390/electronics11071148
Specific Comments:
- The authors mentioned that DRAM accesses were significantly minimized in the proposed design but it is not clear to this reviewer how was it done? Please elaborate on this aspect explicitly and explain it in the paper.
- The proposed deep learning processor was implemented in TSMC 65-nm technology. There is not a single image that demonstrates the chip design, layout and implementation details. Authors need to address this concern and include all necessary details with specific toolchains and associated parameters.
- Could the authors clarify which assembler was used for the proposed ISA implementation?
- Please provide the full details of the toolchain used, the detailed workflow as well as benchmark problems to quantify results. It is all missing at the moment.
- Authors mentioned in their manuscript that the utilization of their design achieved 99%, which is 30%, 6%, and 4% higher than Eyeriss, [20] and [21]. How did the authors measure this? Please specify explicitly.
- The authors mentioned that the work reported in [20] used a 14nm node and they benchmarked with ResNet18 and resNet50 with 1500 MHz frequency and PE utilization of 92 - 98%. The work reported in this paper is very similar (in terms of performance) to the work reported in [20], could the authors elaborate on the differences and advantages of using their proposed ASIP in comparison to [20]?
- The authors analyzed the memory access of several networks such as AlexNet, VGG, and GoogleNet. Authors report that the easiest way is to use the same data reuse and broadcast pattern for all convolutional layers. Could the authors elaborate, on which datasets were used for the benchmarking? please elaborate explicitly.
- Authors report that hybrid multiplexing and broadcasting can minimize memory access. Could you elaborate on the hybrid multiplexing and broadcasting aspect as stated on line 465?
- The scheduling framework was mentioned in several places (line 442, line 465, etc) but you didn’t explain this term explicitly?
- How did the authors measure the energy consumption of the network? please explain explicitly. (line 439)
- Please provide the experimental details and characterization setup including the toolchain used for chip design, characterization and associated parameters. For example, which probe station was used for chip characterization?
Round 2
Reviewer 1 Report
Congrats!
Author Response
Thanks again for your in-depth review and valuable comments.
Reviewer 2 Report
Thanks for making changes to the manuscript.
Please see below the comments that need addressing.
1. Please include Synopsys Design Compiler simulated waveforms to demonstrate the implementation. There is not a single waveform from the simulator in the manuscript.
2. Please mention explicitly in the abstract that the design pipeline was not synthesised, it was rather simulated work.
3. In relation to my previous comment 6, the authors instead of replying and justifying their work with [20], the reply is all related with reference [25]. Could the authors reply to my concern explicitly which is related to [20] rather than [25]?
Round 3
Reviewer 2 Report
Thanks for addressing my comments.
This manuscript is a resubmission of an earlier submission. The following is a list of the peer review reports and author responses from that submission.